# Dietary Factors and the Risk of Depression among Women with Polycystic Ovary Syndrome

**DOI:** 10.3390/nu16060763

**Published:** 2024-03-07

**Authors:** Karolina Łagowska, Joanna Bajerska, Joanna Maria Pieczyńska-Zając

**Affiliations:** Department of Human Nutrition and Dietetics, Poznań University of Life Sciences, Wojska Polskiego 28, 60-637 Poznań, Poland; karolina.lagowska@up.poznan.pl (K.Ł.); joanna.pieczynska@up.poznan.pl (J.M.P.-Z.)

**Keywords:** depression, nutritional habits, polycystic ovary syndrome, obesity

## Abstract

This study aimed to assess the association between dietary factors and depression in a group of polycystic ovary syndrome (PCOS) women and to evaluate potential interactions and the mediating role of BMI in this relationship. One hundred and sixteen women with PCOS were asked to complete the Dietary Habits and Nutrition Beliefs Questionnaire and the manual for developing of nutritional data (KomPAN questionnaire) and the Beck Depression Inventory. The population was divided into two groups: (1) not at risk of depression (ND), *n* = 61, and (2) at risk of depression (RD), *n* = 55. Significantly higher BMI values were observed in the RD group than in the ND group. In the RD group, the intake of vegetables and legumes was lower than in the LD group, but the consumption of sweet beverages and energy drinks was higher. Consumption of vegetables and legumes at least twice per day is known to be associated with a 62% lower probability of the risk of depression in PCOS women. Furthermore, women with overweight and obesity have a 5.82 times greater chance of depression than women with normal body weight. Our findings show that there is a significant association between certain dietary factors, BMI, and symptoms of depression in PCOS women.

## 1. Introduction

Polycystic ovary syndrome (PCOS) is a complex and heterogeneous endocrinopathy among women of premenopausal age, with an estimated prevalence of 5–15% worldwide, using the Rotterdam Consensus criteria [1]. PCOS is commonly characterized by clinical or biochemical hyperandrogenism (excessive synthesis of testosterone or of androstenedione and dehydroepiandrosterone), menstrual disorders (oligo-ovulation, anovulation, amenorrhea, or polycystic ovarian morphology), visceral obesity, and insulin resistance [2]. PCOS may lead to an increased risk of metabolic disorders, including type-2 diabetes mellitus, gestational diabetes, cardiovascular diseases, endometrial cancer, and other pregnancy-related complications. Furthermore, certain characteristics of hyperandrogenism—such as hirsutism, acne, seborrhea, alopecia, and, in extreme cases, lowering of the voice and change in body shape—can significantly diminish the quality of life for women. These symptoms can result in a limiting of social contact, as well as a loss of confidence and serious complexes, leading to depression. Indeed, the prevalence of depression in PCOS is high [3], with a twofold difference in frequency between women with and without PCOS [4].

In the general population, depression affects an estimated 322 million people globally, equivalent to 4.4% of the population, and it is just one of the mental disorders that affect women with PCOS [5]. Women suffering from hirsutism (who make up as many as 60% of women with PCOS) are more likely to exhibit psychotic symptoms, to have higher levels of anxiety and tension, to be more prone to social phobia, and even to have a tendency to suicidal thoughts [6]. Additionally, it has been found that biochemical factors, including elevated circulating testosterone and insulin resistance (typical features of PCOS), have been associated with mental disorders [7]. Moreover, the overall quality of life in women suffering from PCOS is also low, especially among women struggling with infertility [8]. Even global standards emphasize that all PCOS patients should be screened for depression at diagnosis, reflecting growing concern about this aspect of the condition.

Recent data have also indicated the importance of dietary factors in the development of mental disorders in women with PCOS [9]. It should be taken into account that women with PCOS also have an increased risk of eating disorders and, additionally, experience phenomena such as emotional eating, dietary restrictions, and episodes of overeating and compulsive exercise [10]. It has been shown that the occurrence of eating disorders is associated with an increased risk of developing metabolic syndrome [8]. It is well known that metabolic syndrome includes not only “invisible” metabolic disorders, such as hypertension or lipid metabolism disorders, but also central obesity [11,12]. A vicious circle is thus observed because the existence of one elevates the likelihood of developing the other [13].

Little, however, is known about how specific dietary factors are associated with risk of depression in PCOS women. Only a few reports have suggested that high consumption of fried products, on account of their high acrylamide content, may contribute to the development of depression [14], or that fermented foods can alleviate the symptoms of depression [15]. The literature relating to the relevance of particular groups of food products in the treatment or prevention of depression is sparse and ambiguous and has significant limitations. What is more, over the past few years, evidence has emerged about the relationship between nutrition and inflammation with mental disorders such as depression [16,17]. Dietary patterns have been shown to modulate the inflammatory status, thus emphasizing their potential for playing a therapeutic role in depressive disorders [18].

Taking the above into account, this study aimed to assess the association between dietary factors and depression in a group of PCOS women. The potential interactions and the mediating role of BMI in this relationship were also evaluated.

## 2. Materials and Methods

### 2.1. Study Participants

This study enlisted 116 women with PCOS, gathered via recruitment efforts at universities, public venues, and social media platforms. The eligibility criteria for participants included the following:
Having no clinical diagnosis related to eating disorders;Having no clinical diagnosis linked to food allergy or digestive ailments like irritable bowel syndrome, ulcerative colitis, Crohn’s disease, or celiac disease;Not being pregnant and not breastfeeding;Not using contraceptive pills;Not using medications that may affect carbohydrate metabolism;Having a PCOS diagnosis that excludes alternative origins of hyperandrogenism, such as Cushing’s syndrome or hyperprolactinemia.

Participants fulfilled the requirements for PCOS according to the Rotterdam criteria, which are widely used for diagnosing this disorder. These criteria recommend diagnosing PCOS if at least two of the following three characteristics are observed: (1) irregular ovulation; (2) elevated levels of androgens (either clinically or biochemically); and (3) ultrasonographic observations that indicate the presence of a minimum of 12 follicles in each ovary measuring 2–9 mm in diameter or an ovarian volume exceeding 10 mL [19].

### 2.2. Nutritional Habits

To assess dietary habits, we used the validated Dietary Habits and Nutrition Beliefs Questionnaire and the manual for developing of nutritional data (KomPAN questionnaire), which was created by the Behavioral Conditions of Nutrition Team of the Committee for Human Nutrition Science at the Polish Academy of Sciences. The analysis of data from this instrument has been explained in another article [20]. To thoroughly assess dietary quality, two dietary indices were computed:
Pro-Healthy-Diet-Index-10 (pHDI-10, Prohealthy-Diet-Index-10), which is determined by tallying the frequency of intake (number of times per day) of fruit, vegetables, whole (brown)/bread rolls, buckwheat, oats, wholegrain pasta or other coarse-ground groats, milk (including flavored varieties), fermented milk products (like yogurt and kefir), cottage cheese (including processed cheese), fish products and dishes, legume dishes, and meals prepared from white meat (such as chicken, turkey, rabbit, and similar options);Non-Healthy-Diet-Index-14 (nHDI-14, Non-Healthy-Diet-Index-14), determined by totaling the frequency of consumption (number of times per day) of confectionery items, fried foods, alcoholic beverages, sweetened and energy drinks, powdered and instant soups, fast food (such as potato chips, fries, pizza, hot-dogs), energy beverages, white bread and bakery products, “white” grain products (such as white rice, regular pasta, semolina, couscous), butter, animal fat, yellow and blue cheese, and dishes containing red meat (such as veal, mutton, lamb, beef, pork, venison, and smoked sausages).

The questionnaire additionally includes inquiries about physical activity, covering both work- and school-related physical activity and leisure-time physical activity.

The 33 food items in the KomPAN Questionnaire are categorized into 12 food groups for subsequent analysis: (1) products made from refined grains (whitemeal products), (2) products made from whole grains (wholemeal products), (3) dairy products and eggs, (4) various meats and meat-based products, (5) poultry, (6) fish and seafood, (7) fruit, (8) fast food and snacks, (9) confectionary, (10) vegetables and legumes, (11) sweetened beverages, (12) energy drinks, and (13) alcoholic beverages.

### 2.3. The Beck Depression Inventory

The Beck Depression Inventory [BDI] [21] has been described as the most frequently used measure of depression in the literature. The 21-item inventory consists of 4 statements that respondents rate on a scale ranging from 0 to 3, with elevated scores reflecting more advanced depressive symptomatology. The score is the sum of the ratings given by respondents, with a maximum of 63.

It has been suggested to interpret the scores produced by the revised BDI in the following manner:-0–9: absence of depression;-10–18: mild depression;-19–29: moderate depression;-30–63: severe depression.

In this study, the population was divided into two groups: one not at risk of depression (with scores of 0–18: the “ND” group, *n* = 61) and one at risk of depression (with scores of 19–63: the “RD” group, *n* = 55).

### 2.4. Statistical Analysis

Statistical analysis was carried out utilizing Statistica software version 13.0 PL (StatSoft, Tulsa, OK, USA). Data are reported as means and standard deviations (SDs) or as proportions. The normality of distribution of variables was assessed using the Shapiro–Wilk test. Differences in the two groups with their different risks of depression were calculated using independent-sample *t*-tests. For data that were not normally distributed, the Mann–Whitney *U*-test was used. Logistic regression analysis was used to estimate the odds ratios (OR) and the 95% confidence intervals (95% CI) of the estimated BMI, as well as food frequency intake in relation to the risk of depression. To explore the associations between the risk of depression (BDI score) and BMI (kg/m^2^) and frequency intake of specific food groups, we performed multiple logistic regression analyses.

*p-*Values of less than 0.05 were considered statistically significant. The Poznań Medical Ethics Committee (no. 268/18, 8 March 2018) approved this study, and written informed consent was obtained from all patients.

## 3. Results

The characteristics of the subjects recruited to the study are shown in Table 1. In the ND group, 51% of women had no risk of depression (BDI score < 9) and 49% of women had a low risk of depression (BDI score of 10 to 18). In the RD group, 87% of women had moderate risk of depression (with a score of 19–29), while 13% had a severe risk (scoring 30–63). A significant difference in BMI was observed between the ND and RD groups (*p* < 0.001). Additionally, 20% and 14% of the RD group were overweight and obese, respectively. In contrast, in the ND, only 3% were overweight and 6% were obese. In the RD group, the intake of vegetables and legumes was lower, but consumption of sweet beverages and energy drinks was higher than in the ND group. Women in the RD group also had lower pHDI-10 indices (Table 1).

Table 2 shows the effects of BMI and the intake frequency of selected food groups on the risk of depression among women with PCOS. Consumption of vegetables and legumes at least twice per day was associated with a 62% lower probability of risk of depression in PCOS women. Furthermore, pHDI-10 equal to or above the upper quartile also decreased the risk of depression by 57%, while BMI in the case of overweight and obese PCOS women was associated with a risk of depression. However, these correlations were only seen in the crude data: after adjustment, BMI alone appeared as a significant factor associated with the risk of depression in PCOS women. Women with overweight and obesity had a 5.82 times greater chance of depression than women with normal body weight (Table 2).

Multiple logistic regression analysis revealed that BMI (kg/m^2^) is the main predictor of increased risk of depression in PCOS women (Table 3).

## 4. Discussion

The aim of this study was to clarify the associations between BMI, frequency of intake of selected food groups, quality of diet, and risk of depression in women with PCOS.

The BDI results indicate that 47% of PCOS women were at risk of depression, while 6% were at severe risk. McCarron et al. showed that depression is three to eight times more common in women with PCOS than in women without PCOS [22]. In the study of Tan et al., 23.9% of patients had scores indicating mild to moderate depression, while 25.2% had scores indicating clinically relevant depression [23]. Coney et al. showed the median prevalence of depressive symptoms to be 36.6% (interquartile range [IQR]: 22.3–50.0%) in women with PCOS, while the median prevalence for anxiety symptoms was 41.9%. Compared with women without PCOS, these numbers represent a more than threefold increase in the odds of depressive symptoms and a more than fivefold increase in the odds of anxiety symptoms [24]. These results have been confirmed with more recent systematic reviews and meta-analyses [25,26,27]. It is very important to note that the risk of depression persists across ages, with significantly higher prevalence noted in young women and adolescents with PCOS [28]. It has been pointed out that excessive body weight is one of the most important factors associated with the risk of depression in PCOS women [29]. This is also reflected in our study, where as many as 20% of the women in the RD group were overweight, while 14% were obese. In the ND group, only 3% of women were overweight, while 6% were obese. We also found that overweight and obese women with PCOS had an almost 94% greater chance of depression (after adjusting for age). What is more, multiple logistic regression analysis revealed that BMI (kg/m^2^) is associated with increased risk of depression in PCOS women. Maya et al. also noted that increased depression score was significantly correlated with obesity in women with PCOS [30]. Speed et al. have shown that BMI is a causal risk factor for depression, but found no significant evidence that depression causes increased BMI [31]. These findings are consistent with other studies reporting that higher BMI causally increases the risk of depression, but that the inverse relation does not hold [32,33]. Obesity triggers dysregulation of the hypothalamic–pituitary–adrenal (HPA) axis, causing excessive cortisol production. Additionally, cortisol acting via the abundant glucocorticoid receptors in visceral and intra-abdominal regions, which foster central fat deposition and lipid accumulation. Consequently, this process contributes to visceral and central obesity, fostering a detrimental cycle of metabolic dysfunctions. Moreover, insulin resistance is closely linked to depression in women with PCOS [7]. At the molecular level, the underlying cause involves heightened cortisol levels, concomitant with increased sympathetic nerve activity and reduced levels of serotonin (5-HT) within the central nervous system. Furthermore, obese women’s perception of their physical unattractiveness might intensify depressive symptoms and their sense of femininity [34].

Many studies have shown that the daily diet of most women with PCOS is unbalanced and low in dietary fiber, omega-3 fatty acid, calcium, magnesium, zinc, and many vitamins, while also being characterized by excessive amounts of sugar, sodium, total fats, saturated fatty acids, and total cholesterol [35]. Bykowska-Derda et al. have found that PCOS women consume fewer foods with low glycemic indices than do healthy women without PCOS [36]. On the other hand, Zielinska et al. found that dietary nutrients such as folic acid, B-group vitamins, omega-3 fatty acids, zinc, selenium, and magnesium might be associated with the risk of depression in the general population [37]. A recent meta-analysis has also suggested that a high intake of sweetened soft beverages is associated with an increased risk of depression [38]. It should be emphasized that overconsumption of added sugar (such as in energy drinks and sweetened beverages) may elevate the likelihood of depression through various biologically plausible mechanisms. These mechanisms include increased hypothalamic–pituitary–adrenal (HPA) axis reactivity, resulting in the dysregulation of stress response, as well as heightened low-grade inflammation and nonhabituation of the HPA axis induced by obesity [39,40]. This seems to be particularly important in the context of the course of PCOS (which is also regarded as a proinflammatory state) because the chronic increase in inflammatory mediators, such as C-reactive protein and interleukin-6, is closely associated with the pathogenesis of this disease [41]. There is, thus, a possibility that an inflammatory relationship exists between depression and PCOS. It is also possible that the inflammatory markers in PCOS can cross the blood–brain barrier (BBB), leading to the development of depression [41]. Furthermore, Bodnar et al. have indicated that gut microbiota could play a critical role in the signal pathways related to mental health [42], including in in the case of women with PCOS [43]. Unfortunately, the mechanism linking the development of depression in women with PCOS to the altered composition of the intestinal microbiota is still ambiguous [43].

However, mounting evidence indicates that the increasingly popular Western dietary pattern has a disastrous impact on the composition of the intestinal microbiota [44]. It has been noted that rats fed a high-fat Western diet had a significantly greater abundance of *Lachnospiraceae* bacteria than mice fed a standard diet. A positive correlation has also been noted between the abundance of this family of bacteria and the relative size of developed atherosclerotic lesions [45]. Nagpal et al. have also noted that animals consuming a Western diet containing significant amounts of eggs, lard, and fructose had a significantly lower abundance of health-promoting bacteria of the *Lactobacillus* genus, as well as a lower diversity of microbiota, compared to animals fed a Mediterranean diet [46]. It has also been noted that rodents with a higher *Firmicutes:Bacteroidetes* ratio, as induced by a Western diet, reduced exploratory behavior and increased anxiety-like behavior [47]. However, prebiotic supplementation with fructo-oligosaccharides and galacto-oligosaccharides can help reverse unfavorable changes in the intestinal microbiota. The mechanism of action of these prebiotics includes preventing the reduction of beneficial bacteria, such as *Bifidobacterium* and *Lactobacillus*, as well as reducing the concentration of pro-inflammatory cytokines [48]. Indeed, in our study, women in the RD group had lower consumption of vegetables and legumes, which have prebiotic properties. The review by Dharmayani et al. also confirmed that consumption of fruit and vegetables is associated with a decreased risk of developing depression in young people and adults aged 15–45 years [49]. A similar observation was also made by Głąbska et al. [50], who concluded that fruits and vegetables, including some specific subtypes alongside processed fruits and vegetables, seem to have a positive effect on mental health. The general recommendation to consume at least five portions of fruit and vegetables per day may therefore also be beneficial for mental health. It should be pointed out that the lack of such food items, which is typical of Western dietary patterns, might also contribute to gut microbiota dysbiosis and be potentially linked to the greater occurrence of depression [51].

It is also worth noting that, although high levels of fructose in the Western diet are associated with intestinal microbiota disorders, the origin of the fructose is also important: although it is naturally contained in fruit, the amounts are minimal relative to the weight of the fruit itself. Its main sources in the diet are thus fruit juices, as well as nonalcoholic drinks and energy drinks [52]. Moreover, animal studies have shown that the long-term consumption of high amounts of fructose leads to the development of depression. Indeed, the women in our study with PCOS in the RD group had significantly higher intake of fructose sources, such as energy drinks and sweetened beverages.

To the best of our knowledge, our study is one of the first to identify the dietary factors associated with risk of depression in PCOS. Addressing the associations we have identified between nutritional factors, lifestyle factors, and anthropometric parameters is paramount in mitigating the risk of depression among women with PCOS. Additionally, comprehending these associations and implementing a targeted lifestyle or dietary interventions holds promise for averting depression in subsequent generations of women.

Our study has certain limitations. Initially, anthropometric measurements relied on self-reporting, whereas objective methods of body composition measurement like dual-energy X-ray absorptiometry (DXA) are preferable. Secondly, the fact that the questionnaire was completed by the women themselves might have introduced errors due to imperfect memory and potential concealment of inappropriate eating behaviors. However, respondents underwent thorough training before engaging in the survey in order to ensure the collection of accurate nutritional data. To fully understand the treatment of depression in PCOS, a need remains for future large-scale randomized controlled trials. In future investigations, it will be imperative to comprehensively explore the interplay not only between depression and dietary patterns and the history of dietary habits, but also the associations with metabolic or hormonal parameters and gut microbiota composition. What is more, our findings provide justification for future research into the effects and duration of varying diet composition in depression treatment among PCOS women.

## 5. Conclusions

Our findings show a significant association between certain dietary factors, BMI, and the symptoms of depression in PCOS women. Following a healthy diet that is rich in vegetables and legumes, dairy products, and eggs, while intake of energy drinks and sweetened beverages, may lower the risk of developing depression. Future research, as well as medical and dietary advice, should focus on increasing awareness of the nutritional value of the daily diet and its effects on depressive symptoms. It also seems necessary to focus on the composition of the intestinal microbiota and its potential impact on the incidence and course of depression.

## Figures and Tables

**Table 1 nutrients-16-00763-t001:** Characteristics of study participants with food frequency intake values.

	ND (*n* = 61)	RD (*n* = 55)	*p*-Value
Age (years)	27.58 (4.38)	28.74 (5.72)	NS
BMI (kg/m^2^)	23.24 (4.79)	26.86 (6.20)	<0.0001
<18.9 kg/m^2^	2 (6%)	1 (3%)	
19.0–24.9 kg/m^2^	26 (85%)	19 (63)	
25.0–29.9 kg/m^2^	1 (3%)	6 (20%)	
>30 kg/m^2^	2 (6%)	4 (14%)	
Risk of depression (points)	10.23 (3.05)	23.81 (3.81)	<0.0001
<9	31 (51%)	—	
10–18	30 (49%)	—	
19–29	—	48 (87%)	
30–63	—	7 (13%)	
pHDI-10	26.28 (11.18)	21.55 (9.12)	0.014
nHDI-14	20.64 (11.45)	24.12 (13.30)	NS
Whitemeal products (times/day)	0.47 (0.43)	0.52 (0.42)	NS
Wholemeal products (times/day)	0.45 (0.36)	0.37 (0.31)	NS
Dairy products and eggs (times/day)	0.55 (0.40)	0.49 (0.36)	NS
Meat and meat products (times/day)	0.32 (0.40)	0.33 (0.43)	NS
Poultry (times/day)	0.46 (0.48)	0.41 (0.38)	NS
Fish and sea foods (times/day)	0.19 (0.29)	0.22 (0.22)	NS
Fruits (times/day)	0.89 (0.56)	0.91 (0.60)	NS
Fast foods and snacks (times/day)	0.23 (0.21)	0.27 (0.22)	NS
Confectionary (times/day)	0.45 (0.60)	0.47 (0.56)	NS
Vegetable and legumes (times/day)	0.73 (0.47)	0.55 (0.41)	0.03
Sweetened beverages (times/day)	0.14 (0.32)	0.42 (0.63)	0.002
Energy drinks (times/day)	0.11 (0.33)	0.35 (0.59)	0.008
Alcoholic beverages (times/day)	0.12 (0.22)	0.19 (0.28)	NS

ND: no risk or low risk of depression. RD: moderate or severe risk of depression. pHDI-10: Pro-Healthy-Diet-Index-10. nHDI-14: Non-Healthy-Diet-Index-14.

**Table 2 nutrients-16-00763-t002:** Odds ratios (ORs with 95% confidence intervals) of the risk of depression by BMI (kg/m^2^) and intake frequency of selected food groups among PCOS women.

Parameters	Occurrence in Overweight or Obesity Group *n* (%)	Risk of Depression (BDI Score > 19)	
		Crude OR (CI 95%)	OR Adjusted for Age (CI 95%)
BMI (kg/m^2^) overweight or obesity	-	5.92 (2.58; 13.58); *p* < 0.001	5.82 (2.39; 14.21); *p* < 0.001
pHDI-10 ≥upper quartile	13 (28)	0.43 (0.19; 0.97); *p* = 0.039	0.50 (0.14; 1.77); NS
nHDI-14 ≥upper quartile	14 (30)	1.71 (0.68; 4.29); NS	1.04 (0.35; 3.13); NS
Whitemeal products ≥once per day	11 (24)	0.46 (0.16; 1.35); NS	0.43 (0.15; 1.27); NS
Wholemeal products ≥once per day	9 (20)	0.35 (0.10; 1.20); NS	0.35 (0.10; 1.23); NS
Dairy products and eggs ≥once per day	7 (15)	1.22 (0.37; 3.95); NS	1.02 (0.30; 3.43); NS
Meat and meat products ≥two times per week	29 (63)	0.97 (0.39; 2.38); NS	1.38 (0.51; 3.74); NS
Poultry ≥two times per week	38 (83)	1.29 (0.53; 3.14); NS	1.17 (0.46; 2.99); NS
Fish and seafood ≥two times per week	25 (54)	2.03 (0.67; 6.13); NS	2.30 (0.70; 7.60); NS
Fruits ≥two times per day	6 (13)	1.94 (0.62; 6.06); NS	2.04 (0.59; 7.10); NS
Fast foods and snacks ≥two times per week	30 (65)	1.24 (0.48; 3.16); NS	1.03 (0.38; 2.78); NS
Confectionary ≥two times per week	34 (74)	1.31 (0.56; 3.05); NS	1.13 (0.44; 2.89); NS
Vegetable and legumes ≥two times per day	8 (17)	0.38 (0.15; 0.96); 0.04	0.98 (0.25; 3.88); NS
Sweetened beverages ≥two times per week	26 (56)	2.00 (0.59; 6.77); NS	1.73 (0.48; 6.15); NS
Energy drinks ≥once per week	24 (52)	2.33 (0.89; 6.10); NS	1.82 (0.56; 5.93); NS
Alcoholic beverages ≥once per week	15 (33)	1.37 (0.45; 4.20); NS	2.32 (0.88; 6.08); NS

pHDI-10: Pro-Healthy-Diet-Index-10. nHDI-14: Non-Healthy-Diet-Index-14.

**Table 3 nutrients-16-00763-t003:** Associations between age, BMI (kg/m^2^), and intake frequency of specific food groups among PCOS women (regression coefficients with 95% CI).

	Regression Coefficient	95% CI	*p*-Value
Age (years)	0.05	−0.04, 014	0.26
BMI (kg/m^2^)	0.14	0.01, 0.27	0.04
pHDI-10	−0.18	−0.43, 0.08	0.18
nHDI-14	0.00	−0.12, 0.12	0.97
Whitemeal products (times/day)	1.33	−0.28, 2.93	0.11
Wholemeal products (times/day)	−0.10	−1.12, 0.92	0.85
Dairy products and eggs (times/day)	0.45	−2.41, 3.31	0.76
Meat and meat products (times/day)	0.86	−0.93, 2.66	0.35
Poultry (times/day)	0.50	−1.79, 2.79	0.67
Fish and sea foods (times/day)	−1.76	−3.61, 0.10	0.06
Fruits (times/day)	0.47	−2.47, 3.41	0.75
Fast foods and snacks (times/day)	2.01	−1.84, 5.86	0.31
Confectionary (times/day)	−1.39	−4.39, 1.61	0.36
Vegetable and legumes (times/day)	0.85	−0.51, 2.22	0.22
Sweetened beverages (times/day)	0.47	−1.04, 1.99	0.54
Energy drinks (times/day)	−0.36	−2.57, 1.86	0.75
Alcoholic beverages (times/day)	2.11	−0.14, 4.35	0.07

pHDI-10: Pro-Healthy-Diet-Index-10. nHDI-14: Non-Healthy-Diet-Index-14.

## Data Availability

The data presented in this study are available on reasonable request from the first author, Karolina Łagowska (karolina.lagowska@up.poznan.pl). The data are not publicly available.

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
