# Peer review of "Dietary Factors and the Risk of Depression among Women with Polycystic Ovary Syndrome"

_nutrients, 2024, doi:10.3390/nu16060763_

Round 1
Reviewer 1 Report
Comments and Suggestions for Authors
The study aimed to assess the association between dietary factors and depression in a group of PCOS women while controlling for the potential mediating effect of BMI.
A) Regarding the presentation of the results, I would like to point out the following:
On page 3 line 130 it is stated that in the ND group 27% of women have no risk of depression (BDI < 9) and 26% have a low risk of depression (BDI=10-18). These two percentages do not add up to 100% of the group (which consists only of women at no risk or at low risk of depression). Something is wrong with the reported rates. Please check.
The same applies to the percentages of the RD group. It is stated in line 131 that 63% of the women in the group have a moderate risk of depression and 9% a severe risk. Here again, the two percentages do not add up to 100%.
Furthermore in Table 1 the rates reported for the two groups of women are completely different (variable Risk of Depression). Authors should carefully and diligently check what is mentioned in the final form of their article.
In line 137, LD group should become ND group.
In Table 1, it is necessary that the percentages refer to the total number of individuals in each group and not to the total number of individuals in the study sample.
Also in Table 1, the units in which the mean values (and the SD) of the food groups are calculated must be mentioned E.g. 0.47 for Whitemeal products refers to what units?
B) Regarding the methodological part of the study, I have to point out the following:
While the study concerns the differences between the two compared groups (women at risk for depression, women not at risk) in terms of their eating habits, the final regression model on which the results are based has the BDI score as the dependent variable. This approach does not give the odds ratios for each group of consuming more or less of certain food groups. To move the analysis in the correct direction, a logistic regression model must be run, where the response variable will be the two groups and the predictors will be the food groups and, in addition, the BMI and the age of the women (as possible mediators in the relationship between diet and depression). The model will be run with the Enter method and the results table will show all the predictors of the model whether they are significant or not. (One of our primary concerns in multivariate analysis is to point estimate the coefficients of all predictors in the model, mutually adjusted for each other).
Comments on the Quality of English Language
Minor editing of English language would improve the text.
Author Response
Author’s response to Reviews
Manuscript: Dietary factors and the risk of depression among women with Polycystic Ovary Syndrome
Date: February 2024
Authors:
Karolina Łagowska (karolina.lagowska@up.poznan.pl)
Joanna Bajerska (joanna.bajerska@up.poznan.pl)
Joanna Maria Pieczyńska-Zając (joanna.pieczynska@up.poznan.pl)
Reviewer 1
General Authors answer: Thank you for your opinion and all your suggestions. Authors have made all the corrections in the paper.
On page 3 line 130 it is stated that in the ND group 27% of women have no risk of depression (BDI < 9) and 26% have a low risk of depression (BDI=10-18). These two percentages do not add up to 100% of the group (which consists only of women at no risk or at low risk of depression). Something is wrong with the reported rates. Please check. The same applies to the percentages of the RD group. It is stated in line 131 that 63% of the women in the group have a moderate risk of depression and 9% a severe risk. Here again, the two percentages do not add up to 100%. Furthermore in Table 1 the rates reported for the two groups of women are completely different (variable Risk of Depression). Authors should carefully and diligently check what is mentioned in the final form of their article.
Authors answer: Authors corrected the mistakes and now we these data are more transparent and readable.
Line 152-158: The characteristics of the subjects recruited to the study are shown in Table 1. In the ND group, 51% of women had no risk of depression (BDI score < 9) and 49% of women had a low risk of depression (BDI score of 10 to 18). In the RD group, 87% of women had moderate risk of depression (with a score of 19–29) while 13% had a severe risk (scoring 30–63). A significant difference in BMI was observed between the ND and RD groups (p < 0.001). Additionally, 20% and 14% of the RD group were overweight and obese, respectively. In contrast, in the ND only 3% were overweight and 6% obese.
In line 137, LD group should become ND group.
Authors answer: Authors corrected a mistake and now we have ND instead of LD.
Line 158-161: In the RD group, the intake of vegetable and legumes was lower, but consumption of sweet beverages and energy drinks was higher than in the ND group. Women in the RD group also had lower pHDI-10 indices (Table 1).
In Table 1, it is necessary that the percentages refer to the total number of individuals in each group and not to the total number of individuals in the study sample.
Authors answer: Authors corrected the mistakes and now we these data percentages refer to the total number of individuals in each group.
Table 1. Characteristics of study participants with food frequency intake values.
|
ND n = 61 |
RD (n = 55) |
p-value |
Age (years) |
27.58 (4.38) |
28.74 (5.72) |
NS |
BMI (kg/m2) < 18.9 kg/m2 19.0–24.9 kg/m2 25.0–29.9 kg/m2 > 30 kg/m2 |
23.24 (4.79) 2 (6%) 26 (85%) 1 (3%) 2 (6%) |
26.86 (6.20) 1 (3%) 19 (63) 6 (20%) 4 (14%) |
< 0.0001 |
Risk of depression (points) < 9 10–18 19–29 30–63 |
10.23 (3.05) 31 (51%) 30 (49%) — — |
23.81 (3.81) — — 48 (87%) 7 (13%) |
< 0.0001 |
Also in Table 1, the units in which the mean values (and the SD) of the food groups are calculated must be mentioned E.g. 0.47 for Whitemeal products refers to what units?
Authors answer: Authors added necessary information (frequency of consumption (times/day)) to the table 1.
Table 1. Characteristics of study participants with food frequency intake values.
|
ND n = 61 |
RD (n = 55) |
p-value |
Age (years) |
27.58 (4.38) |
28.74 (5.72) |
NS |
BMI (kg/m2 < 18.9 kg/2 19.0–24.9kg/m2 25.0–29.9 kg/m2 > 30 kg/m2 |
23.24 (4.79) 2 (6%) 26 (85%) 1 (3%) 2 (6%) |
26.86 (6.20) 1 (3%) 19 (63) 6 (20%) 4 (14%) |
< 0.0001 |
Risk of depression (points) < 9 10–18 19–29 30–63 |
10.23 (3.05) 31 (51%) 30 (49%) — — |
23.81 (3.81) — — 48 (87%) 7 (13%) |
< 0.0001 |
pHDI-10 |
26.28 (11.18) |
21.55 (9.12) |
0.014 |
nHDI-14 |
20.64 (11.45) |
24.12 (13.30) |
NS |
Whitemeal products (times/day) |
0.47 (0.43) |
0.52 (0.42) |
NS |
Wholemeal products (times/day) |
0.45 (0.36) |
0.37 (0.31) |
NS |
Dairy products and eggs (times/day) |
0.55 (0.40) |
0.49 (0.36) |
NS |
Meat and meat products (times/day) |
0.32 (0.40) |
0.33 (0.43) |
NS |
Poultry (times/day) |
0.46 (0.48) |
0.41 (0.38) |
NS |
Fish and sea foods (times/day) |
0.19 (0.29) |
0.22 (0.22) |
NS |
Fruits (times/day) |
0.89 (0.56) |
0.91 (0.60) |
NS |
Fast foods and snacks (times/day) |
0.23 (0.21) |
0.27 (0.22) |
NS |
Confectionary (times/day) |
0.45 (0.60) |
0.47 (0.56) |
NS |
Vegetable and legumes (times/day) |
0.73 (0.47) |
0.55 (0.41) |
0.03 |
Sweetened beverages (times/day) |
0.14 (0.32) |
0.42 (0.63) |
0.002 |
Energy drinks (times/day) |
0.11 (0.33) |
0.35 (0.59) |
0.008 |
Alcoholic beverages (times/day) |
0.12 (0.22) |
0.19 (0.28) |
NS |
While the study concerns the differences between the two compared groups (women at risk for depression, women not at risk) in terms of their eating habits, the final regression model on which the results are based has the BDI score as the dependent variable. This approach does not give the odds ratios for each group of consuming more or less of certain food groups. To move the analysis in the correct direction, a logistic regression model must be run, where the response variable will be the two groups and the predictors will be the food groups and, in addition, the BMI and the age of the women (as possible mediators in the relationship between diet and depression). The model will be run with the Enter method and the results table will show all the predictors of the model whether they are significant or not. (One of our primary concerns in multivariate analysis is to point estimate the coefficients of all predictors in the model, mutually adjusted for each other).
Authors answer: According to the reviewer suggestions authors improved the analysis connected with logistic regression model.
Line 179-186: Stepwise multiple linear regression analysis revealed that vegetables and legumes are the main predictors of reduced risk of depression in the ND group (Table 3). On the other hand, stepwise multiple linear regression analysis revealed in the RD group that sweetened drinks were the main predictors of the risk of depression (Table 4).
Table 3. Multiple stepwise linear regression of the risk of depression (BDI score) with BMI (kg/m2) and intake frequency of specific food groups in PCOS women in the ND group
Variables |
Regression β coefficient |
Std Error β |
p-value |
Model for the risk of depression (score) Adjusted R2 = 0.224 |
|||
Vegetable and legumes (times/day) |
-0.330 |
0.928 |
0.024 |
Fruits (times/day) |
0.249 |
0.742 |
0.064 |
Dairy products and eggs (times/day) |
-0.243 |
0.946 |
0.051 |
Whitemeal products (times/day) |
0.211 |
1.494 |
0.117 |
Table 4. Multiple stepwise linear regression of the risk of depression (BDI score) with BMI (kg/m2) and intake frequency of specific food groups in PCOS women in the RD group
Variables |
Regression β coefficient |
Std Error β |
p-value |
Model for the risk of depression (score) Adjusted R2 = 0.363 |
|||
Sweetened beverages (times/day) |
0.373 |
0.945 |
0.009 |
Vegetable and legumes (times/day) |
-0.172 |
1.372 |
0.197 |
BMI (kg/m2) |
0.239 |
0.094 |
0.080 |
Fruits (times/day) |
-0.217 |
1.035 |
0.142 |
Dairy products and eggs (times/day) |
-0.155 |
1.574 |
0.245 |
Reviewer 2
General Authors answer: Thank you for your opinion and all your suggestions. Authors have made all the corrections in the paper.
In discussion method, please pay more attention to the relationship of cause and result of different risk factors and PCOS. e.g. PCOS lead to obesity and had a close loop back to depression? Or obesity had a close cause to the food choice preference.
Authors answer: Thank you for this suggestion. In the discussion section there was many information connected with relationship of cause and result of different risk factors and PCOS. But Authors add necessary other information.
Line 239-245: This seems to be particularly important in the context of the course of PCOS (which is also regarded as a proinflammatory state) because the chronic increase in inflammatory mediators, such as C-reactive protein and in-terleukin-6, is closely associated with the pathogenesis of this disease [41]. There is thus a possibility that an inflammatory relationship exists between depression and PCOS. It is also possible that the inflammatory markers in PCOS can cross the blood-brain barrier (BBB), leading to the development of depression [41].
The article can add some limitation or future plan in the conclusion part.
Authors answer: In the paper we had some information about limitations. But according to the reviewers suggestion we added some necessary information about future investigation.
Line 292-304: Our study has certain limitations. Initially, anthropometric measurements relied on self-reporting, whereas objective methods of body composition measurement like dual-energy X-ray absorptiometry (DXA) are preferable. Secondly, the fact that the questionnaire was completed by the women themselves might have introduced errors due to imperfect memory and potential concealment of inappropriate eating behaviors. However, respondents underwent thorough training before engaging in the survey in order to ensure the collection of accurate nutritional data. To fully understand the treatment of depression in PCOS, a need remains for future large-scale randomized controlled trials. In future investigations, it will be imperative to comprehensively explore the interplay not only between depression and dietary patterns and the history of dietary habits, but also the associations with metabolic or hormonal parameters and gut microbiota composition. What is more, our findings provide justification for future research into the effects and duration of varying diet composition in depression treatment among PCOS women.

Reviewer 2 Report
Comments and Suggestions for Authors
In discussion method, please pay more attention to the relationship of cause and result of different risk factors and PCOS.
e.g. PCOS lead to obesity and had a close loop back to depression? Or obesity had a close cause to the food choice preference.
The article can add some limitation or future plan in the conclusion part.
Author Response

(The authors gave the same response as above.)

Round 2
Reviewer 1 Report
Comments and Suggestions for Authors
Dear authors, thank you for the corrections you made to your article following my review. But, as I mentioned in my first response, your study has the design of a case-control study. So the only way to compare the eating habits of the two groups of women is to use a multiple logistic regression model in the way I indicated. To move the analysis in the correct direction, a logistic regression must be run, with response variable the two groups of women and predictors the food groups. In addition, the BMI and the age of the women must be controlled as possible mediators in the analysis. The logistic model will be run with the Enter method and the results table will show all the predictors (food groups, BMI and age) of the model whether they are significant or not.
Author Response
Manuscript: Dietary factors and the risk of depression among women with polycystic ovary syndrome
Date: March 2024
Authors:
Karolina Łagowska (karolina.lagowska@up.poznan.pl)
Joanna Bajerska (joanna.bajerska@up.poznan.pl)
Joanna Maria Pieczyńska-Zając (joanna.pieczynska@up.poznan.pl)
Reviewer 1
Dear authors, thank you for the corrections you made to your article following my review. But, as I mentioned in my first response, your study has the design of a case-control study. So the only way to compare the eating habits of the two groups of women is to use a multiple logistic regression model in the way I indicated. To move the analysis in the correct direction, a logistic regression must be run, with response variable the two groups of women and predictors the food groups. In addition, the BMI and the age of the women must be controlled as possible mediators in the analysis. The logistic model will be run with the Enter method and the results table will show all the predictors (food groups, BMI and age) of the model whether they are significant or not.
Authors answer: Thank you for your opinion and your suggestion. According to the reviewer suggestions authors improved the analysis connected with logistic regression model.
Table 3. Associations between age, BMI (kg/m2) and intake frequency of specific food groups among PCOS women (regression coefficients with 95% CI).
|
Regression coefficient |
95% CI |
p-value |
Age (years) |
0.05 |
-0.04, 014 |
0.26 |
BMI (kg/m2) |
0.14 |
0.01, 0.27 |
0.04 |
pHDI-10 |
-0.18 |
-0.43, 0.08 |
0.18 |
nHDI-14 |
0.00 |
-0.12, 0.12 |
0.97 |
Whitemeal products (times/day) |
1.33 |
-0.28, 2.93 |
0.11 |
Wholemeal products (times/day) |
-0.10 |
-1.12, 0.92 |
0.85 |
Dairy products and eggs (times/day) |
0.45 |
-2.41, 3.31 |
0.76 |
Meat and meat products (times/day) |
0.86 |
-0.93, 2.66 |
0.35 |
Poultry (times/day) |
0.50 |
-1.79, 2.79 |
0.67 |
Fish and sea foods (times/day) |
-1.76 |
-3.61, 0.10 |
0.06 |
Fruits (times/day) |
0.47 |
-2.47, 3.41 |
0.75 |
Fast foods and snacks (times/day) |
2.01 |
-1.84, 5.86 |
0.31 |
Confectionary (times/day) |
-1.39 |
-4.39, 1.61 |
0.36 |
Vegetable and legumes (times/day) |
0.85 |
-0.51, 2.22 |
0.22 |
Sweetened beverages (times/day) |
0.47 |
-1.04, 1.99 |
0.54 |
Energy drinks (times/day) |
-0.36 |
-2.57, 1.86 |
0.75 |
Alcoholic beverages (times/day) |
2.11 |
-0.14, 4.35 |
0.07 |
pHDI-10: Pro-Healthy-Diet-Index-10
nHDI-14: Non-Healthy-Diet-Index-14
